# ANALOGXPERT: AUTOMATING ANALOG TOPOLOGY SYNTHESIS BY INCORPORATING CIRCUIT DESIGN EXPERTISE INTO LARGE LANGUAGE MODELS

## ABSTRACT

Analog circuits are crucial in modern electronic systems, and automating their design has attracted significant research interest. One of major challenges is *topology synthesis*, which determines circuit components and their connections. Recent studies explore large language models (LLM) for topology synthesis. However, the scenarios addressed by these studies do not align well with practical applications. Specifically, existing work uses vague design requirements as input and outputs an ideal model, but detailed structural requirements and device-level models are more practical. Moreover, current approaches either formulate topology synthesis as graph generation or Python code generation, whereas practical topology design is a complex process that demands extensive design knowledge. In this work, we propose AnalogXpert, a LLM-based agent aiming at solving practical topology synthesis problem by incorporating circuit design expertise into LLMs. First, we represent analog topology as SPICE code and introduce a subcircuit library to reduce the design space, in the same manner as experienced designers. Second, we decompose the problem into two sub-task (i.e., block selection and block connection) through the use of CoT and in-context learning techniques, to mimic the practical design process. Third, we introduce a proofreading strategy that allows LLMs to incrementally correct the errors in the initial design, akin to human designers who iteratively check and adjust the initial topology design to ensure accuracy. Finally, we construct a high-quality benchmark containing both real data (30) and synthetic data (2k). AnalogXpert achieves 40% and 23% success rates on the synthetic dataset and real dataset respectively, which is markedly better than those of GPT-4o (3% on both the synthetic dataset and the real dataset).

## 1 INTRODUCTION

Analog circuits form the backbone of many modern electronic systems, playing a crucial role in processing continuous signals to achieve a variety of tasks in the devices that permeate our everyday lives. The significance of analog circuits lies in their ability to excel in situations where information must be captured and processed directly from natural sources, such as sound Zhu & Feng (2024), light Muramatsu et al. (2003), temperature Wang et al. (2013), or pressure Wang et al. (2017). Analog circuit design usually can be divided into three stages: **(1) Topology synthesis.** Zhao & Zhang (b) Topology synthesis determines the whole analog circuit topology. This stage will select the basic devices (e.g. transistors, capacitors, resistors) and give the connection relationship among these basic devices. **(2) Circuit Sizing.** Xiaohan Gao (2024); Wang et al. (2020); Chen et al. (2022); Cao et al. (2022) The selected basic devices need to be applied with the appropriate parameters in order to maximize the circuit performance for a given topology. **(3) Layout synthesis.** Chen et al.; Zhang et al. (2024; 2023); Dhar et al. This stage performs the placement and routing for an analog circuit to generate the final layout.

Analog topology synthesis is the foundation of the entire circuit design and determines the performance of the circuit. Given the importance of analog topology synthesis, some research has emerged and focused on the automation of it. Early work explores utilize graph neural networks (GNN) and reinforcement learning (RL) (Zhao & Zhang, b;c; Dong et al.) for topology synthesis. Recently, as large language models (LLMs) have shown impressive capabilities across various fields, there has

Table 1: **Comparison of AnalogXpert and related works.** AnalogXpert leverages a prompt-based LLM agent to generate the analog circuit topology. With the subcircuit library-based design space reduction method, AnalogXpert can reach the design target more easily. The final benchmark for AnalogXpert includes both real-world analog cases and synthetic datasets which makes the result more convincing.

| Method | Design Space Reduction Method[1] | Benchmark Type | Primary Methodology |
|---|---|---|---|
| CKTGNN Dong et al. | Ideal Model | synthetic | Graph Neural Network |
| RLATS Zhao & Zhang (b) | Subcircuit Library | real | Policy Gradient Neural Network |
| LADAC Liu et al. (a) | None | real | Prompt-based LLM |
| LaMAGIC Chang et al. | Ideal Model | synthetic | SFT-based LLM |
| Artisan Zihao Chen (2024) | Ideal Model | real | SFT-based LLM |
| AnalogCoder Lai et al. (a) | None | real | Prompt-based LLM |
| **AnalogXpert** | Subcircuit Library | real+synthetic | Prompt-based LLM |

[1] Method to simplify the analog circuits design task.

been increased interest in leveraging them for topology synthesis tasks (Wei et al., 2023; Chen et al., 2024; Yao et al., 2023; Renze & Guven, 2024; Xu et al., 2024). Although significant progress has been made, the problem of topology synthesis remains largely unresolved. One issue is that their focused scenarios are not aligned closely with practical applications. Specifically, they take vague design specifications as input, e.g., 'cascode current mirror'. Instead, more precise structure design requirements, like the number of stages and the need for compensation, are more practical and beneficial in real applications. Additionally, most approaches focus on ideal models, which fails to cover the entire legalized design space and cannot be directly converted into the final topology. Another issue is that previous studies either treat the topology synthesis purely as graph generation (Chang et al.) or python code generation (Lai et al., a). However, in practice, topology synthesis is a complex process requiring extensive circuit design experience and domain-specific knowledge.

In this work, we introduce AnalogXpert, designed to address more practical analog topology synthesis problem while utilizing circuit design expertise for optimal performance. As shown in Figure 1, AnalogXpert takes detailed structure design requirements as input and output a real circuit topology design (rather than ideal models), which is represented as *SPICE code*. The intuition behind is that the SPICE code is widely used in practical design scenarios and exists on the internet which is very likely to have been seen in the pre-training of LLMs. Therefore, formulating analog topology as SPICE code not only matches the practical scenarios but also facilitates understanding by LLMs.

Based on the above formulation, AnalogXpert presents a LLM-based agent for topology synthesis by incorporating circuit design expertise into LLMs (see Figure 2). First, a well-designed subcircuit library is proposed to reduce the design space and improve the generation success rate. With the customized text-format subcircuit library, AnalogXpert can generate the final topology from the subcircuit level rather than the device level which not only aligns with human design practices but also greatly reduces the length of the model output. Second, the design task decomposition is performed to make the design more logical and easy to check for errors. AnalogXpert leverages the CoT to prompt LLMs to generate topology step by step. Specifically, the design task is decomposed into block selection and block connection just as humans do in practical design scenarios. Third, analog designers usually need to check the block types and connection relationships after they finish the circuit design. This approach greatly improves the accuracy of analog topology generation. AnalogXpert also takes this into account and proposes a proofreading strategy to check the generated analog topologies and give revision messages back to LLMs for iterative refinement.

Since the task formulation of AnalogXpert is very different from previous works, we construct a new benchmark including both real data (30) and synthetic data (2k). Previous studies only have one type of data, either real data or synthetic data. Meanwhile, the number of real data is very limited ($\leq 5$), except for the AnalogCoder Lai et al. (a) (24). Validated on the proposed benchmark, AnalogXpert achieves 40% and 23% success rates in the synthetic dataset and real dataset respectively, which is much better than the GPT-4o (3% in the synthetic dataset, 3% in the real dataset). The experimental results demonstrate the effectiveness and robustness of the AnalogXpert.

The main **contributions** of this paper can be summarized as: (1) We focus on a more pracitcal topology synthesis problem. We formulate it as a SPICE code generation problem and introduce a design space reduction method based on an extensible subcircuit library. (2) We propose a CoT-based LLM agent to imitate the human design process, decomposing topology synthesis tasks into subcircuit

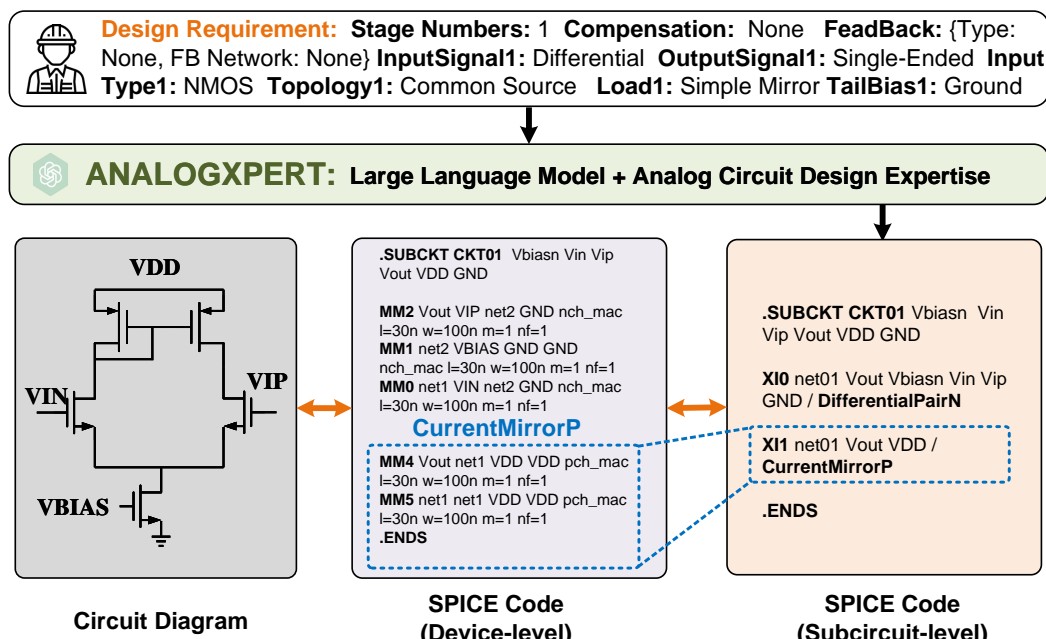

Figure 1: AnalogXpert formulates the topology synthesis task as SPCIE code generation.

block selection and connection graph construction. (3) We propose a proofreading strategy based on the human experience, which makes LLMs revise the generated topology iteratively, to further improve the design ability of LLM agents. With several rounds of self-refinement, LLM agents can avoid some basic mistakes and improve the design success rate. (4) We also propose a holistic benchmark to completely validate the design ability of AnalogXpert. The proposed benchmark consists of 30 real-world analog design tasks and 2k synthetic data.

## 2 RELATED WORK

### 2.1 ANALOG TOPOLOGY SYNTHESIS AUTOMATION

Analog topology synthesis is the most challenging step in the analog design flow and thus has attracted extensive research interest. Before the introduction of LLMs, some research has already attempted to automate analog topology synthesis with versatile AI methods. For example, RLATS Zhao & Zhang (b;c) leverages reinforcement learning(RL) to build up the analog circuit step by step. RLATS establishes a subcircuit library to simplify the topology synthesis problem so that the RL agent is able to handle it. However, the design diversity of analog circuits makes it difficult to transfer RL agents between different circuit types. CKTGNN Dong et al. builds up a 10k dataset to train a graph neural network which achieves an impressive performance. The drawback is that CKTGNN leverages the ideal model to reduce the design space. The ideal model can not represent the entire legalized design space and can not be directly converted to the final topology. With the introduction of LLM, more research works on the automation of topology synthesis have emerged. LADAC Liu et al. (a) and Artisan Zihao Chen (2024) leverage prompting and supervised fine-tuning(SFT) to enhance the analog design ability of LLM, respectively. However, they only validate the proposed framework on a few (≤5) real analog cases. This is not enough to demonstrate that LLM has adequate analog circuit design capabilities. LaMAGIC Chang et al. build up a 120k synthetic dataset to support the SFT of LLM, but its design tasks are simpler than the actual analog design tasks. LaMAGIC actually focuses on a kind of radio frequency circuit with limited basic devices (≤ 6) including a capacitor(2 terminals), inductor(2 terminals), and switch(2 terminals). Analog usually is made up of tens of basic devices (0∼50), such as transistors (4 terminals), capacitors (2 terminals), and resistors (2 terminals). AnalogCoderLai et al. (a) leverages a prompt-based LLM to generate the analog topology and validate the framework on some real data. The disadvantage is that user requirements are very ambitious and can be handled directly by existing LLM agents

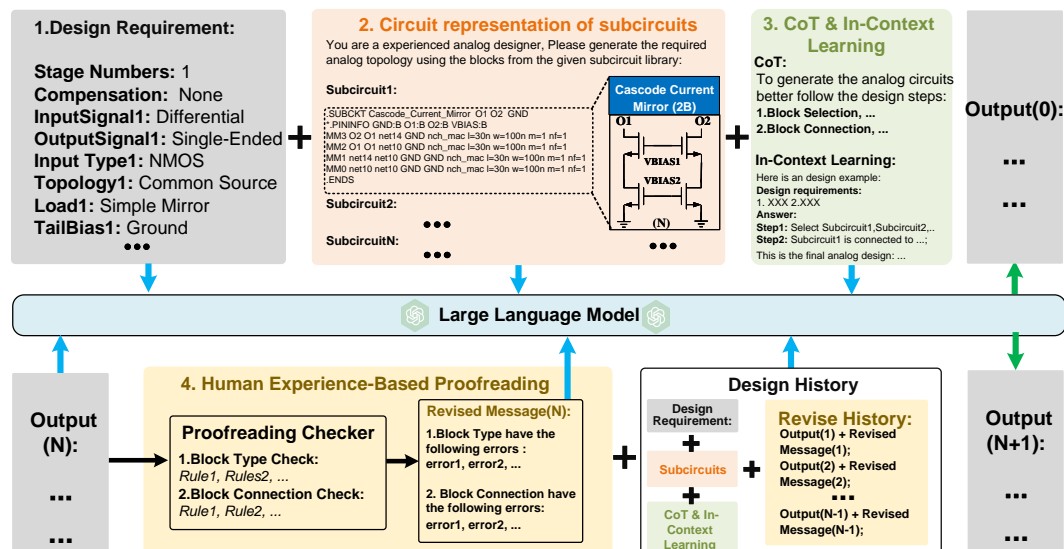

Figure 2: AnalogXpert takes versatile design requirements of analog circuit structures as input, such as stage number, input signal type, feedback type and so on. AnalogXpert performs the topology synthesis based on given subcircuit blocks that are SPICE code format for easy use by LLM agents. The proposed CoT first selects the subcircuit blocks and then determines block connection relationships. Meanwhile, a corresponding design example are provided for the in-context learning. After obtain the initial results, the circuit design will be verified by a proofreading checker in terms of block types and block connections. If there is an error in the circuit design then a revised message will be generated by the checker. The revised message as well as the previous generation history are given to the LLM agents and the next result will be generated.

in most cases. In real-world scenarios, designers would like to give more detailed design conditions. Therefore, we propose AnalogXpert to process concrete conditions and validate the framework on both real data and synthetic data which is a more comprehensive demonstration of LLM's analog design capabilities

## 2.2 LARGE LANGUAGE MODELS

Large language models with pre-trained parameters such as GPT, LLaMa Dubey et al. (2024); Touvron et al. (2023), Claude, have demonstrated terrific ability in versatile tasks. Recent research on prompting has further enhanced the LLM ability. For example, Chain-of-Thought(CoT) Wei et al. (2023) is a technique that encourages language models to generate intermediate reasoning steps in a step-by-step manner, rather than directly providing the final answer. This method improves the model's ability to solve complex tasks by making its reasoning process more transparent and structured. Tree-of-Thought(ToT) Yao et al. (2023) is a famous ToT evolution of CoT, which allows multiple possible solutions or reasoning paths to be explored simultaneously. This approach helps improve decision-making in complex tasks by evaluating different branches of thought before converging on a final answer. Some other researches focus on In-context learning Dong et al. (2024) which involves giving a language model examples of tasks or instructions within the same prompt to improve the final performance. In addition to pre-generation promptings, post-generation feedback is equally important. Some feedback techniques such as self-reflection Renze & Guven (2024); Madaan et al. (2023), can also improve the LLM performance. These techniques provide the foundation for constructing agents with practical functions, such as gene-editing Huang et al., math Ahn et al. (2024), and chip designs Liu et al. (b); Blocklove et al.; Lai et al. (b); He et al.. Based on these techniques, we introduce the circuit design expertise, which allows LLM to really deal with analog topology synthesis problem.

## 3 METHOD

**Method Overview.** In this section, we elaborate on the detailed methods of AnalogXpert (as Figure 2 shows), an efficient training-free analog design agent incorporating human design expertise. AnalogXpert introduces a brand new task formulation of conditional analog topology synthesis which can further demonstrate the design ability of LLM agents (Section 3.1). Leveraging a well-designed subcircuit library, AnalogXpert constructs a novel and effective analog circuit representation which greatly helps LLM agents design analog circuits concisely (Section 3.2). A domain-specific prompting design flow based on CoT is proposed and further enhanced by the in-context learning (Section 3.3). Although the methods mentioned above can improve the design capability of LLM agents, the single-round generation mode still struggles to handle complex tasks. Therefore, AnalogXpert further introduces human experience-based proofreading to help LLM agents gradually correct the mistakes and finally generate the correct topology in several rounds (Section 3.4). With the cooperation of these techniques, AnalogXpert has a decent performance in the conditional analog topology synthesis tasks.

### 3.1 PROBLEM FORMULATION

We propose a brand new formulation of analog topology synthesis as Equation 1 shows. We formulates the topology synthesis as a real-world SPICE code generation problem. The devices $\{D_i\}$ and there connection relationships $\{\sum_{n=1}^{N} N_j(D_i, T_k)_n\}$ are represented directly in the real world SPICE code. Our formulation is very different from ideal model Dong et al.; Chang et al.; Zihao Chen (2024) and Python code-based formulation Lai et al. (a). For ideal model formulations, they have different types of basic devices $\{D_i\}$ and smaller $N$ indicates less connection relationships. For Python code-based formulation, the $\{D_i\}$ and $\{\sum_{n=1}^{N} N_j(D_i, T_k)_n\}$ are represented in Python code which requires an extra step to convert to real SPICE code.

$$\{D_i\}, \{\sum_{n=1}^{N} N_j(D_i, T_k)_n\} = F\{Ckt, \sum_{n=1}^{N} Struc_n\} \quad (1)$$

Apart from the format, the main difference between AnalogXpert and previous work lies in the input conditions. Previous work takes design specifications $\sum_{n=1}^{N} Spec_n$ as input to generate the final circuit topology. There are two disadvantages of previous method: (1)The input condition ($\sum_{n=1}^{N} Spec_n$) is ambiguous. A series of required design specifications can be satisfied with many different topologies. At the same time, one topology may also satisfy different specifications. (2)The input condition ($\sum_{n=1}^{N} Spec_n$) includes complex mathematical calculations. The relationship between the analog topology and design specifications is often described by some complex mathematical equations which is difficult and inappropriate for LLM agents to deal with. Therefore, AnalogXpert leverages some structure requirements $\sum_{n=1}^{N} Struc_n$ to instead the specifications $\sum_{n=1}^{N} Spec_n$. Structure requirements directly describe the characteristics of the analog topology which are more concrete than the specifications. Meanwhile, structure requirements have successfully separated the complex mathematical calculations from the generation tasks. Such formulation turns the analog topology synthesis problem into a pure sequence-to-sequence problem which is more appropriate for LLM agents.

### 3.2 ANALOG CIRCUIT REPRESENTATION

The conventional SPICE code is built up from devices, such an approach leads to a flexible but complex generation task which is hard for LLM agents to deal with. In the real analog design process, the analog designers often use the subcircuits instead of the devices. Inspired by this, we propose a novel subciruit-level SPICE code representation that is built up from subcircuits. For example (as Figure 3 shows), devices01-03 belongs to the same subcircuit01 and thus can be simplified to one line. The details of our subcircuit library are shown at the bottom of Figure 3, including one-signal path blocks, two-signal path blocks, capacitors, and resistors. Figure 3 only shows some important part of the library, the complete subcircuit library is detailed in Appendix A.2. It is important to note that the subcircuit library is summarized from analog design experience and can be extended easily.

To summarize a high-quality subcircuit library, we basically refer to subcircuit libraries from related studies Meissner & Hedrich; Zhao & Zhang (a;b) in electronic design automation(EDA) fields. Moreover, we refer to some analog circuit design books Razavi to make minor modifications of the subcircuit library aiming to be more practical.

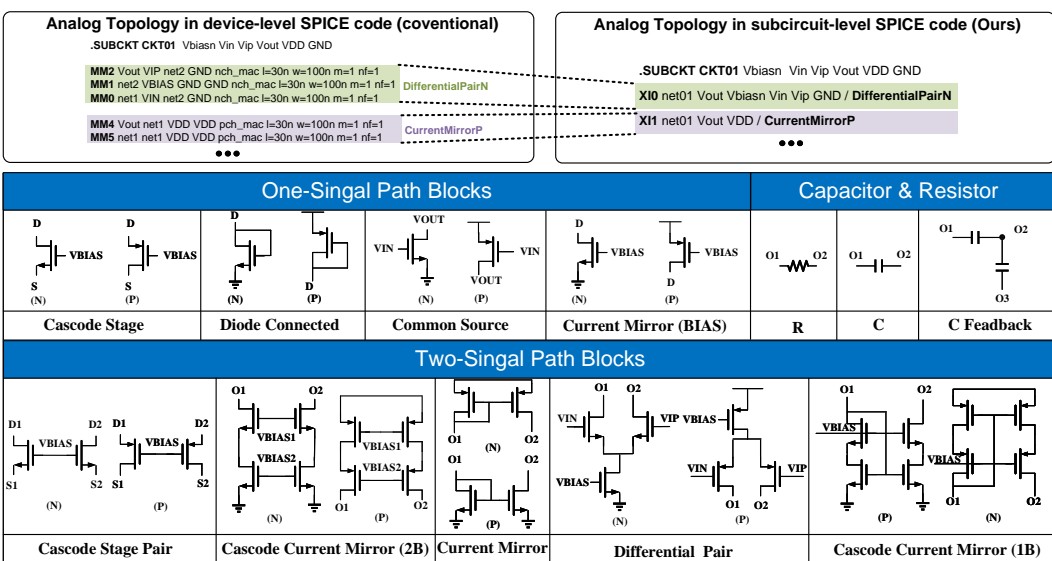

Figure 3: We propose the subcircuit-level SPICE code representation for analog topology. This subcircuit library can be easily extended to support versatile application scenarios.

### 3.3 DESIGN TASK DECOMPOSITION

AnalogXpert mainly leverages CoT and in-context learning. The CoT has two major steps: **(1) Block Selection.** Based on the previous subcircuit library, the analog topology synthesis task can be performed like the human design process. In this step, LLM agents select the appropriate subcircuits from the library according to the design requirements. **(2) Block Connection.** With the selected subcircuits, LLM agents then determine the connection relationships and generate the final analog topology. For in-context learning, AnalogXpert selects some design task examples as prompts. The principle of selection is the similarity of design requirements. For a given design requirement, AnalogXpert will give the most similar design task as an example. With the CoT and in-context learning, the basic generation functions can be achieved.

### 3.4 HUMAN EXPERIENCE-BASED PROOFREADING

During the generation of analog circuit topology, we actually expect the LLM agent to follow some design rules. A straightforward idea is that we summarize these rules as text and then use them as prompts. However, when dealing with a series of rules, the prompt gets longer and longer, and it becomes difficult for an LLM agent to fully understand these rules and follow them strictly. Therefore, we propose human experience-based proofreading to get LLM agents out of this dilemma. The basic idea of proofreading is depicted in Figure 4. In practice, the initial netlist does not provide enough information for the rule-based checker. We annotate these netlists with the terminal types including current output (I/O), current input (I/I), and voltage (V). In this way, the previous subcircuit library can be summarized into 10 types, making the error-checking process simpler (shown as the right-top part of Figure 4). After the annotation, a proofreading checker will detect the corresponding errors, which are mainly categorized into block selection errors and block connection errors. For the accuracy of checking, the proofreading checker is implemented with deterministic programs rather than LLM agents. The detected errors as well as the violated rules make up the final refinement prompt. The LLM agents take the refinement prompt to generate the refined analog topology. AnalogXpert will repeat this process until the design meets the design requirements or reaches the maximum iterations. It is worth noting that not only the current refinement tips are provided, but also the generation and refinement history in order to avoid making similar mistakes as much as possible.

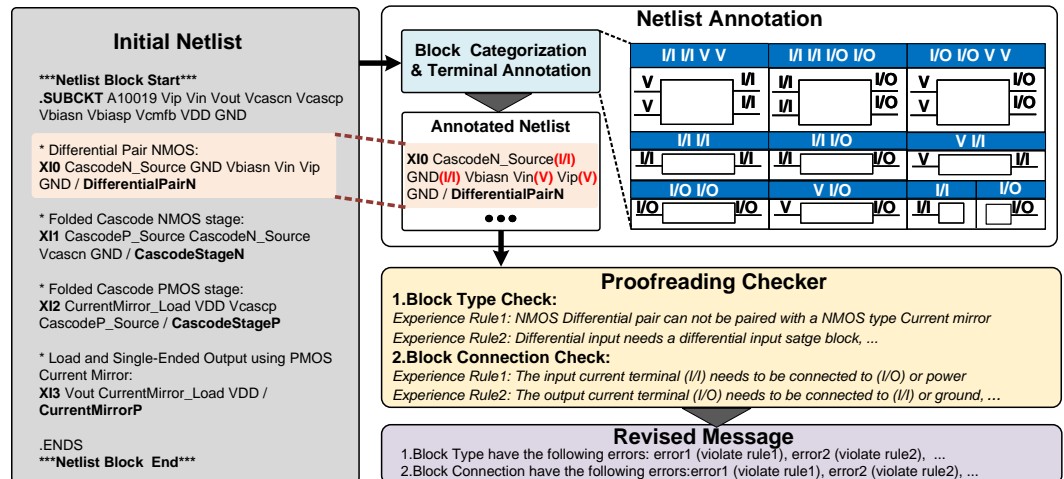

Figure 4: Human experience-based proofreading encodes the experience rules in an external checker. With the assisted information provided by annotation, the external checker can detect the circuit design errors and provide the refinement prompt.

## 4 EXPERIMENTS

**Baseline.** We compare AnalogXpert with pure GPT-3.5 and GPT-4o, which are advanced models with strong code generation ability and rich cross-disciplinary knowledge. For other work related to topology synthesis, as they handle different problems with us, the comparison with them is infeasible and thus is excluded. Specifically, AnalogXpert tackles concrete structure design requirement while most related work explores ambiguous design specifications (Dong et al.; Zhao & Zhang, b; Liu et al., a; Chang et al.; Zihao Chen, 2024; Lai et al., a). Besides, AnalogXpert directly works on real-world device-level model and SPICE code while the related work focuses on ideal model (Dong et al.; Chang et al.; Zihao Chen, 2024) and Python code-based format (Lai et al., a). If these methods are used for our tasks, the results will be close to pure LLMs (e.g. GPT-4o, GPT-3.5) performance (as listed in Table 2) due to the different task formulation.

**Benchmark.** We construct two benchmarks for the final evaluation, including a real data benchmark and a synthetic data benchmark. The real data benchmark is collected from a commercial tool named AnalogDesignToolbox Anonymous. There are approximately 60 different analog design topologies, and we select the most representative 30 analog topologies as the real data benchmark. The synthetic data benchmark is built by a random generation Python code leveraging the subcircuit library. Each synthetic data consists of four parts, the stage number, the input blocks, other given blocks, and the maximum number of blocks. The generation task on synthetic data is selecting some subcircuit blocks based on the input blocks and other given blocks to form the final circuit design. The total number of used blocks should be less than the maximum number of blocks. The stage is set from one to three, the input blocks and other blocks are randomly selected from the proposed subcircuit library, and the maximum number of blocks is randomly selected from a range related to the stage number(e.g. for one stage the range is 2-5). Finally, we generate 2k synthetic data with totally different structure design requirements.

**Metrics.** The metric of AnalogXpert is whether the LLM agent can generate the correct analog topology for the given design requirements in one trial. For real data, the correctness can only be determined if all blocks and connections exactly match the design requirements. Such strict correctness requires humans to check the analog topology results directly. For synthetic data, the block selection and connection have some basic rules to follow. If a generated circuit topology does not violate these rules, it will be determined as correct. Such correctness can be checked by an automatic program. Tested on a certain number of cases, the correct ratio can be obtained. The correct ratio can directly reflect the ability of the LLM agent to generate the required analog topology.

Table 2: **Main results.** Four different methods are validated, including the pure **GPT-3.5 Turbo**, GPT-3.5 Turbo with our proposed methods(**GPT-3.5Turbo+Ours**), the pure **GPT-4o**, and the **AnalogXpert** implemented on the GPT-4o.

| Model TaskId | GPT-3.5Turbo | | GPT-3.5Turbo+Ours | | GPT-4o | | AnalogXpert | |
|---|---|---|---|---|---|---|---|---|
| | statistic | correct ratio | statistic | correct ratio | statistic | correct ratio | statistic | correct ratio |
| Synthetic Data | | | | | | | | |
| Task1 (one stage) | 1/10 | 10% | 5/20 | 25% | 2/10 | 20% | 16/20 | **80%** |
| Task2 (two stage) | 0/15 | 0% | 41/125 | 33% | 1/15 | 13% | 53/125 | **42%** |
| Task3 (three stage) | 0/20 | 0% | 156/625 | 25% | 0/20 | 0% | 221/625 | **35%** |
| Task4 (one stage one component) | 0/20 | 0% | 60/150 | 40% | 0/20 | 0% | 86/150 | **57%** |
| Task5 (two stage one component) | 0/20 | 0% | 234/750 | 31% | 0/20 | 0% | 327/750 | **44%** |
| Task6 (three stage one component) | 0/15 | 0% | 66/330 | 20% | 0/15 | 0% | 98/330 | **30%** |
| Average | 1/100 | 1% | 562/2000 | 28% | 3/100 | 3% | 801/2000 | **40%** |
| Real Data | | | | | | | | |
| Real Task | 1/30 | 3% | 0/30 | 0% | 1/30 | 3% | 7/30 | **23%** |

Table 3: **Abliation Study.** The ablation on AnalogXpert components and proofreading rounds are conducted.

| Method TaskId | WoT CoT&In context | | WoT Proofreading | | AnalogXpert(1R) | | AnalogXpert(5R) | | AnalogXpert(10R) | |
|---|---|---|---|---|---|---|---|---|---|---|
| | statistic | correct ratio | statistic | correct ratio | statistic | correct ratio | statistic | correct ratio | statistic | correct ratio |
| Task1 (one stage) | 7/20 | 35% | 4/20 | 20% | 7/20 | 35% | 15/20 | 75% | 16/20 | **80%** |
| Task2 (two stage) | 31/125 | 25% | 6/125 | 5% | 10/125 | 8% | 39/125 | 31% | 53/125 | **42%** |
| Task3 (three stage) | 162/625 | 26% | 29/625 | 5% | 48/625 | 8% | 122/625 | 0% | 221/625 | **35%** |
| Task4 (one stage one component) | 68/150 | 45% | 31/150 | 21% | 32/150 | 21% | 66/150 | 44% | 86/150 | **57%** |
| Task5 (two stage one component) | 266/750 | 35% | 64/750 | 9% | 110/750 | 15% | 224/750 | 30% | 327/750 | **44%** |
| Task6 (three stage one component) | 95/330 | 29% | 15/330 | 5% | 19/330 | 6% | 54/330 | 16% | 98/330 | **30%** |
| Average | 629/2000 | 31% | 149/2000 | 7% | 226/2000 | 11% | 520/2000 | 26% | 801/2000 | **40%** |

**Main Results.** In the main experiment, each method conducts seven different tasks including six tasks on synthetic data and one task on the real data. The design requirements of Task 1-3 have different input stage numbers (1-3) without any given blocks. The design requirements of Task 4-6 have different input stage numbers (1-3) with an extra given block. The pure GPT-3.5 Turbo and GPT-4o are only tested on one hundred synthetic data because the automatic check program can not be performed without the subcircuit library-based representation. Thus, the results are checked by humans. On synthetic data Pure GPT-4o and pure GPT-3.5 Turbo can only achieve a correct ratio of 3% and 1%, respectively. Such experimental results demonstrate the importance of the proposed subcircuit library-based representation method. We can also observe that AnalogXpert(40%) outperforms GPT-3.5Turbo+Ours(28%) on synthetic data. This result indicates that the proposed framework needs models with sufficient comprehension to generate the topology more accurately. On real data, only AnalogXpert achieves a 23% correct ratio, other methods have near-zero correct ratios. The failure of GPT-3.5Turbo+Ours on real data is due to poor model comprehension and a significant increase in task difficulty. The experimental results on both benchmarks demonstrate the effectiveness of the proposed AnalogXpert in dealing with complex connection relationships and real-world topology synthesis problems.

**Ablation Study.** We perform the ablation study on each component of AnalogXpert except for the subcircuit library-based representation. Subcircuit library-based representation is the foundation of CoT and proofreading and the framework will become pure GPT-4o without it. The experimental results indicate that CoT & In-context learning has less impact on performance compared to the proofreading strategy. We also conduct the experiments with different proofreading rounds. The experimental results are also consistent with the intuition that the correct ratio increases as the number of proofreading rounds increases, proving the effectiveness of this strategy.

**Visulization.** Three visualized results of the real data are shown in Figure 5. Each case shows the failed circuit design in the generation process and is then corrected by AnalogXpert itself during the proofreading step. The failure reason in case 1 is the connection error due to the floating current

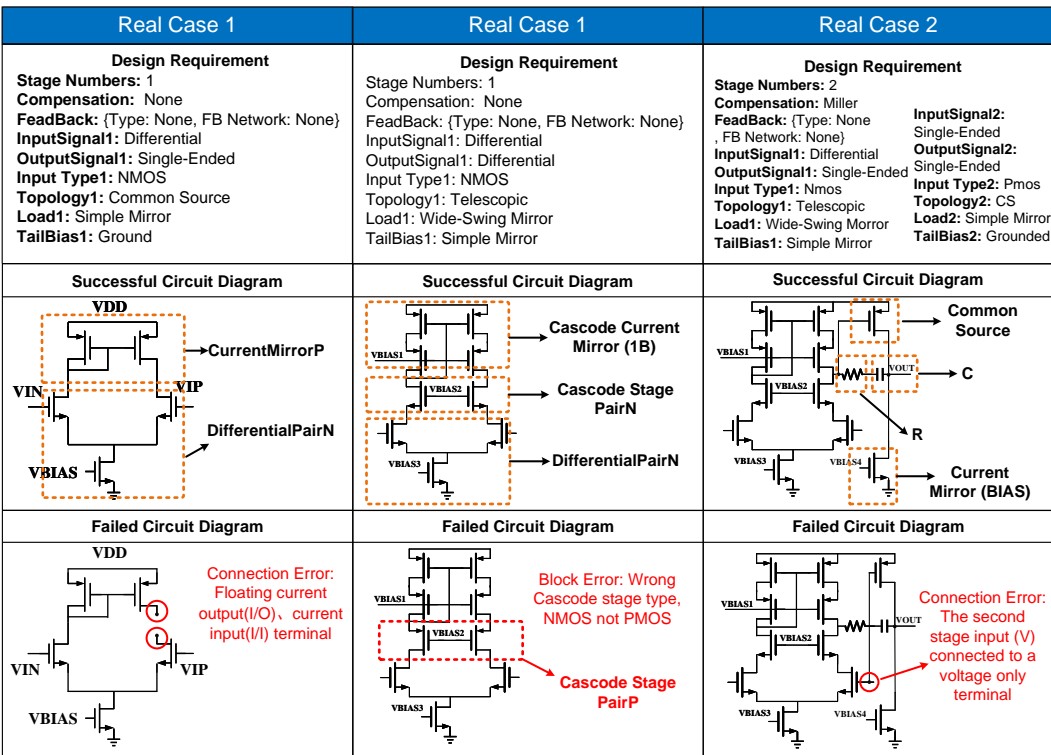

Figure 5: Three real cases with successful circuit diagrams and failed circuit diagrams.

input and output terminal which is not allowed. In real case 2, the AnalogXpert makes a mistake in the selection of subcircuits. the cascode stage should be N-type but AnalogXpert first selects the P-type. Real case3 is a two-stage amplifier, the AnalogXpert connects the input of the second stage to the input of the first stage which destroys the structure of the two stages.

## 5 CONCLUSION

In this work, we propose AnalogXpert, a powerful analog topology synthesis tool based on training-free LLMs. AnalogXpert leverages the subcircuit-based circuit representation, CoT&in-context learning, and human experience-based proofreading to imitate the human design process and improve the design accuracy. Both real data and synthetic data benchmarks are constructed on the structural requirements tasks. The experimental results (40% and 23% correct ratio in real and synthetic data) demonstrate the effectiveness of the AnalogXpert.

**Limitation and Future Directions.** Although this work has constructed both a real dataset and a synthetic dataset, the size of the real dataset is still small. A larger real dataset may lead to a brand new prompting method in the analog topology synthesis problem which can achieve a better correct ratio. In the future, AnalogXpert can be used to generate sufficient synthetic data of the analog circuit topologies in the SPICE code format. With high-quality synthetic data, some small models can be fine-tuned to achieve a competitive performance. Having fine-tuned mini-models that can be run locally ensures the security of design data, which is important for commercial circuit design companies.

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

## A APPENDIX

### A.1 PROMPTS OF THE ANALOGXPERT

The prompt of real tasks are provided as follows.

---

## User (Design prompt for real tasks)

You are a professional analog designer, and now you need to design the required analog circuits with the given design libaray of some analog basic components. Here is the libaray details, including Cell NAME, PinINFO and detail Description:

**[Subcirtuit Library Prompt]**

1. Cell Name: CascodeStageN
   PININFO: DRAIN(I/I) SOURCE(I/O) VBIAS(B) GND(P)
   Description: Single NMOS Cascode

2. Cell Name: CascodeStageNPair
   PININFO: DRAIN1(I/I) SOURCE1(I/O) DRAIN2(I/I) SOURCE2(I/O) VBIAS(B) GND(P)
   Description: A Pair of NMOS Cascode

3. Cell Name: CascodeStageP
   PININFO: DRAIN(I/O) SOURCE(I/I) VBIAS(B) VDD(P)
   Description: Single PMOS Cascode

4. Cell Name: CascodeStagePPair
   PININFO: DRAIN1(I/O) SOURCE1(I/I) DRAIN2(I/O) SOURCE2(I/I) VBIAS(B) VDD(P)
   Description: A Pair of PMOS Cascode

5. Cell Name: DiodeConnectedN
   PININFO: DRAIN(I/I) GND(P)
   Description: DiodeConnected Signle NMOS

6. Cell Name: DiodeConnectedP
   PININFO: DRAIN(I/O) VDD(P)
   Description: DiodeConnected Signle PMOS

7. Cell Name: CommonSourceN
   PININFO: DRAIN(I/I) VIN(V) GND(P)
   Description: common source single NMOS amplifier

8. Cell Name: CommonSourceP
   PININFO: DRAIN(I/O) VIN(V) VDD(P)
   Description: common source single PMOS amplifier

9. Cell Name: CurrentMirrorCSN
   PININFO: O1(I/I) O2(I/I) VBIAS(B) GND(P)

---

Description: Cascode Current Mirror based on NMOS with single bias

10. Cell Name: CurrentMirrorCSN_B
    PININFO: O1(I/I) O2(I/I) VBIAS1(B) VBIAS2(B) GND(P)
    Description: Cascode Current Mirror based on NMOS with two seperate bias

11. Cell Name: CurrentMirrorCSN_S
    PININFO: O1(I/I) O2(I/I) GND(P)
    Description: Cascode Current Mirror based on NMOS without seperate bias, two self connected current mirror

12. Cell Name: CurrentMirrorCSP
    PININFO: O1(I/O) O2(I/O) VBIAS(B) VDD(P)
    Description: Cascode Current Mirror based on PMOS with single bias

13. Cell Name: CurrentMirrorCSP_B
    PININFO: O1(I/O) O2(I/O) VBIAS1(B) VBIAS2(B) VDD(P)
    Description: Cascode Current Mirror based on PMOS with two seperate bias

14. Cell Name: CurrentMirrorCSP_S
    PININFO: O1(I/O) O2(I/O) VDD(P)
    Description: Cascode Current Mirror based on PMOS without seperate bias, two self connected current mirror

15. Cell Name: CurrentMirrorN
    PININFO: O1(I/I) O2(I/I) GND(P)
    Description: Simple Current Mirror based on NMOS

16. Cell Name: CurrentMirrorP
    PININFO: O1(I/O) O2(I/O) VDD(P)
    Description: Simple Current Mirror based on PMOS

17. Cell Name: CurrentSourceN
    PININFO: DRAIN(I/I) VBIAS(B) GND(P)
    Description: Bias Control Current Source based on NMOS

18. Cell Name: CurrentSourceP
    PININFO: DRAIN(I/O) VBIAS(B) GND(P)
    Description: Bias Control Current Source based on PMOS

19. Cell Name: DifferentialPairN
    PININFO: O1(I/I) O2(I/I) VBIAS(B) VIN(V) VIP(V) GND(P)
    Description: Differential Pair based on NMOS, with tail bias

20. Cell Name: DifferentialPairP
    PININFO: O1(I/O) O2(I/O) VBIAS(B) VIN(V) VIP(V) VDD(P)
    Description: Differential Pair based on PMOS, with tail bias

21. Cell Name: DifferentialPairPBS
    PININFO: O1(I/O) O2(I/O) VBIAS(B) VIN(V) VIP(V) VDD(P)
    Description: Differential Pair based on PMOS (Bulk connected to Source), with tail bias

22. Cell Name: R
    PININFO: O1(P) O2(P)
    Description: Resistor

23. Cell Name: C
    PININFO: O1(V) O2(V)
    Description: Capacitor

24. Cell Name: Cap_Feadback
    PININFO: Vout(V) Vin(V) mid(V)
    Description: Feadback nets with two Capacitors

**[Experience rule related Prompt]**

Apart from this there are also some basic things you should konw:
1.The TailBias1 have been included in the DifferentialPairN/P/PBS subcircuit, you needn't to set the tail bias seperately. But you should check that only Differential Input need the simple mirror TailBias.

2.Most times, the PMOS input may matches the NMOS current mirror. So, usually please don't match the PMOS input with a PMOS current mirror.

3.Don't use the MOFFET and R/C directly, use the subcircuits 1-23

**[CoT Task Decomposition Prompt]**

For the generation step, Please following these steps:

1. Step1: According to the stage number and then select the appropriate basic components from the library for each stage;

2. Step2: Connect the select blocks to form the final ceicuits. Note that the current flow ports(I/O) must be matched by corresponding current inflow ports(I/I) or (P) during the connection process.Generate the final netlist, the netlist should Start with '''***Netlist Start***''' and end with '''***Netlist End***''', and the terminal type should also be listed including(I/I),(I/O),(V),(B),(P); (V) is the vin/vout; (B) is the bias; (P) is the power;

**[In-context Learning Prompt]**

There will be some specific user qurries and you should given the result that satisfied the user qurries. Here is an example:

**User Query:**
Stage Numbers: 1
Compensation: None

FeadBack: Type: Inverting, FB Network: Capacitive
InputSignal1: Differential
OutputSignal1: Differential
Input Type1: NMOS
Topology1: Common source
Load1: Simple Mirror
TailBias1: Simple Mirror

**Answer:**
***Netlist Start***
.SUBCKT A10018 Vin Vip Voutn Voutp Vbiasn Vbiasp VDD GND
XI0 Voutn(I/I) Voutp(I/I) Vbiasn(V) Vipm(V) Vinm(V) GND(V) / DifferentialPairN
XI1 Voutn(I/O) VDD(I/I) Voutp(I/O) VDD(I/I) Vbiasp(V) VDD(V) / CascodeStagePPair
XI2 Voutn(V) Vip(V) Vipm(V) / Cap_Feadback
XI2 Voutp(V) Vin(V) Vinm(V) / Cap_Feadback
.ENDS
***Netlist End***

The prompt of synthetic tasks are provided as follows.

## User (Design prompt for synthetic tasks)

You are a professional analog designer, and now you need to design the required analog circuits with the given design libaray of some analog basic components. Here is the libaray details, including Cell NAME, PinINFO and detail Description:

**[Subcirtuit Library Prompt]**
*** Inputblocks Start ***

1. PinInfo: T1(I/I) T2(V)
   CellName: CommonSourceN

2. PinInfo: T1(I/O) T2(V)
   CellName: CommonSourceP

3. PinInfo: T1(I/O) T2(I/O) T3(V) T4(V)
   CellName: DifferentialPairP DifferentialPairPBS

4. PinInfo: T1(I/I) T2(I/I) T3(V) T4(V)
   CellName: DifferentialPairN

\*\*\* Inputblocks End \*\*\*

\*\*\* Otherblocks Start \*\*\*

1. PinInfo: T1(I/I) T2(I/O)
   CellName: CascodeStageN CascodeStageP

2. PinInfo: T1(I/I) T2(I/I) T3(I/O) T4(I/O)
   CellName: CascodeStageNPair CascodeStagePPair

3. PinInfo: T1(I/I) T2(I/I)
   CellName: CurrentMirrorCSN CurrentMirrorCSN_B CurrentMirrorCSN_S CurrentMirrorN

4. PinInfo: T1(I/O) T2(I/O)
   CellName: CurrentMirrorCSP CurrentMirrorCSP_B CurrentMirrorCSP_S CurrentMirrorP

5. PinInfo: T1(I/I)
   CellName: CurrentSourceN DiodeConnectedN

6. PinInfo: T2(I/O)
   CellName: CurrentSourceP DiodeConnectedP

\*\*\* Otherblocks End \*\*\*

**[CoT Task Decomposition Prompt]**

For each userquerry, you should follow the given Inputblocks and the given Otherblocks and select some blocks from other blocks under the limitation of max blocks number. To do this better please following these steps:

1. Step1: Select a number of Otherblocks to form the circuit up to the given maximum value–max blocks number

2. Step2: The current flow ports(I/O) must be matched by corresponding current inflow ports(I/I) during the connection process. Generate the final netlist, the netlist should Start with '''\*\*\*Netlist Start\*\*\*''' and end with '''\*\*\*Netlist End\*\*\*''', and the terminal type should also be listed including(I/I),(I/O),(V); you should use "net1,net2,net3" and so on to represent the net name, do not use other names.

[In-context Learning Prompt]

Here is an example:

**Userquerry:**
Input Stage: 1
Input blocks: DifferentialPairP
Other blocks:
Max blocks number: 2

**Response:**
***Netlist Start***
DifferentialPairP / net01(I—O) net02(I—O)
CurrentMirrorCSN / net01(I—I) net02(I—I)
***Netlist End***

## A.2 THE SPICE CODE OF THE SUBCIRCUIT LIBRARY

The detailed SPICE code of the proposed subcircuit library is shown as follows.

### SPICE code of the Subcircuit Library

```
*************************************************************************
*Library Name:  DATASET *Cell Name:  CascodeStageN *View Name:  schematic
*************************************************************************

.SUBCKT CascodeStageN DRAIN SOURCE VBIAS GND
*.PININFO DRAIN:B GND:B SOURCE:B VBIAS:B
MM0 DRAIN VBIAS SOURCE GND nch_mac l=30n w=100n m=1 nf=1
.ENDS

*************************************************************************
*Library Name:  DATASET *Cell Name:  CascodeStageNPair *View Name:  schematic
*************************************************************************
.SUBCKT CascodeStageN DRAIN1 SOURCE1 DRAIN2 SOURCE2 VBIAS GND
*.PININFO DRAIN:B GND:B SOURCE:B VBIAS:B
MM0 DRAIN1 VBIAS SOURCE1 GND nch_mac l=30n w=100n m=1 nf=1
MM1 DRAIN2 VBIAS SOURCE2 GND nch_mac l=30n w=100n m=1 nf=1
.ENDS

*************************************************************************
*Library Name:  DATASET *Cell Name:  CascodeStageP *View Name:  schematic
*************************************************************************
.SUBCKT CascodeStageP DRAIN SOURCE VBIAS VDD
*.PININFO DRAIN:B SOURCE:B VBIAS:B VDD:B
MM0 DRAIN VBIAS SOURCE VDD pch_mac l=30n w=100n m=1 nf=1
.ENDS
```

```
**************************************************************************
*Library Name: DATASET *Cell Name: CascodeStagePPair *View Name: schematic
**************************************************************************
.SUBCKT CascodeStageP DRAIN1 SOURCE1 DRAIN2 SOURCE2 VBIAS VDD
*.PININFO DRAIN:B SOURCE:B VBIAS:B VDD:B
MM0 DRAIN VBIAS SOURCE VDD pch_mac l=30n w=100n m=1 nf=1
MM1 DRAIN VBIAS SOURCE VDD pch_mac l=30n w=100n m=1 nf=1
.ENDS

**************************************************************************
*Library Name: DATASET *Cell Name: DiodeConnectedN *View Name: schematic
**************************************************************************
.SUBCKT DiodeConnectedN DRAIN GND
*.PININFO DRAIN:B GND:B VIN:B
MM0 DRAIN DRAIN GND GND nch_mac l=30n w=100n m=1 nf=1
.ENDS

**************************************************************************
*Library Name: DATASET *Cell Name: DiodeConnectedP *View Name: schematic
**************************************************************************
.SUBCKT CommonSourceN DRAIN VDD
*.PININFO DRAIN:B GND:B VIN:B
MM0 DRAIN DRAIN VDD VDD nch_mac l=30n w=100n m=1 nf=1
.ENDS

**************************************************************************
*Library Name: DATASET *Cell Name: CommonSourceN *View Name: schematic
**************************************************************************
.SUBCKT CommonSourceN DRAIN VIN GND
*.PININFO DRAIN:B GND:B VIN:B
MM0 DRAIN VIN GND GND nch_mac l=30n w=100n m=1 nf=1
.ENDS

**************************************************************************
*Library Name: DATASET *Cell Name: CommonSourceP *View Name: schematic
**************************************************************************
.SUBCKT CommonSourceP DRAIN VIN VDD
*.PININFO DRAIN:B VDD:B VIN:B
MM0 DRAIN VIN VDD VDD pch_mac l=30n w=100n m=1 nf=1
.ENDS

**************************************************************************
*Library Name: DATASET *Cell Name: CurrentMirrorCSN *View Name: schematic
**************************************************************************
.SUBCKT CurrentMirrorCSN O1 O2 VBIAS GND
*.PININFO GND:B O1:B O2:B VBIAS:B
MM3 O2 VBIAS net14 GND nch_mac l=30n w=100n m=1 nf=1
MM2 O1 VBIAS net10 GND nch_mac l=30n w=100n m=1 nf=1
MM1 net14 O1 GND GND nch_mac l=30n w=100n m=1 nf=1
MM0 net10 O1 GND GND nch_mac l=30n w=100n m=1 nf=1
.ENDS

**************************************************************************
*Library Name: DATASET *Cell Name: CurrentMirrorCSNB *View Name: schematic
**************************************************************************
.SUBCKT CurrentMirrorCSNB O1 O2 VBIAS1 VBIAS2 GND
*.PININFO GND:B O1:B O2:B VBIAS:B
```

```
MM3 O2 VBIAS1 net14 GND nch_mac l=30n w=100n m=1 nf=1
MM2 O1 VBIAS1 net10 GND nch_mac l=30n w=100n m=1 nf=1
MM1 net14 VBIAS2 GND GND nch_mac l=30n w=100n m=1 nf=1
MM0 net10 VBIAS2 GND GND nch_mac l=30n w=100n m=1 nf=1
.ENDS

*******************************************************************
*Library Name: DATASET *Cell Name: CurrentMirrorCSNBS *View Name: schematic
*******************************************************************
.SUBCKT CurrentMirrorCSNBS O1 O2 GND
*.PININFO GND:B O1:B O2:B VBIAS:B
MM3 O2 O1 net14 GND nch_mac l=30n w=100n m=1 nf=1
MM2 O1 O1 net10 GND nch_mac l=30n w=100n m=1 nf=1
MM1 net14 net10 GND GND nch_mac l=30n w=100n m=1 nf=1
MM0 net10 net10 GND GND nch_mac l=30n w=100n m=1 nf=1
.ENDS

*******************************************************************
*Library Name: DATASET *Cell Name: CurrentMirrorCSP *View Name: schematic
*******************************************************************
.SUBCKT CurrentMirrorCSP O1 O2 VBIAS VDD
*.PININFO O1:B O2:B VBIAS:B VDD:B
MM3 net14 O1 VDD VDD pch_mac l=30n w=100n m=1 nf=1
MM2 net15 O1 VDD VDD pch_mac l=30n w=100n m=1 nf=1
MM5 O2 VBIAS net14 VDD pch_mac l=30n w=100n m=1 nf=1
MM4 O1 VBIAS net15 VDD pch_mac l=30n w=100n m=1 nf=1
.ENDS

*******************************************************************
*Library Name: DATASET *Cell Name: CurrentMirrorCSPB *View Name: schematic
*******************************************************************
.SUBCKT CurrentMirrorCSPB O1 O2 VBIAS1 VBIAS2 VDD
*.PININFO O1:B O2:B VBIAS:B VDD:B
MM3 net14 VBIAS2 VDD VDD pch_mac l=30n w=100n m=1 nf=1
MM2 net15 VBIAS2 VDD VDD pch_mac l=30n w=100n m=1 nf=1
MM5 O2 VBIAS1 net14 VDD pch_mac l=30n w=100n m=1 nf=1
MM4 O1 VBIAS1 net15 VDD pch_mac l=30n w=100n m=1 nf=1
.ENDS

*******************************************************************
*Library Name: DATASET *Cell Name: CurrentMirrorCSPBS *View Name: schematic
*******************************************************************
.SUBCKT CurrentMirrorCSPBS O1 O2 VDD
*.PININFO O1:B O2:B VBIAS:B VDD:B
MM3 net14 net15 VDD VDD pch_mac l=30n w=100n m=1 nf=1
MM2 net15 net15 VDD VDD pch_mac l=30n w=100n m=1 nf=1
MM5 O2 O1 net14 VDD pch_mac l=30n w=100n m=1 nf=1
MM4 O1 O1 net15 VDD pch_mac l=30n w=100n m=1 nf=1
.ENDS

*******************************************************************
*Library Name: DATASET *Cell Name: CurrentMirrorN *View Name: schematic
*******************************************************************
.SUBCKT CurrentMirrorN O1 O2 GND
*.PININFO GND:B O1:B O2:B
MM1 O2 O1 GND GND nch_mac l=30n w=100n m=1 nf=1
MM0 O1 O1 GND GND nch_mac l=30n w=100n m=1 nf=1
```

```
.ENDS

*****************************************************************
*Library Name:  DATASET *Cell Name:  CurrentMirrorP *View Name:  schematic
*****************************************************************
.SUBCKT CurrentMirrorP O1 O2 VDD
*.PININFO O1:B O2:B VDD:B
MM3 O2 O1 VDD VDD pch_mac l=30n w=100n m=1 nf=1
MM2 O1 O1 VDD VDD pch_mac l=30n w=100n m=1 nf=1
.ENDS

*****************************************************************
*Library Name:  DATASET *Cell Name:  CurrentSourceN *View Name:  schematic
*****************************************************************
.SUBCKT CurrentSourceN DRAIN VBIAS GND
*.PININFO DRAIN:B GND:B VBIAS:B
MM0 DRAIN VBIAS GND GND nch_mac l=30n w=100n m=1 nf=1
.ENDS

*****************************************************************
*Library Name:  DATASET *Cell Name:  CurrentSourceP *View Name:  schematic
*****************************************************************
.SUBCKT CurrentSourceP DRAIN VBIAS VDD
*.PININFO DRAIN:B VBIAS:B VDD:B
MM1 DRAIN VBIAS VDD VDD pch_mac l=30n w=100n m=1 nf=1
.ENDS

*****************************************************************
*Library Name:  DATASET *Cell Name:  DifferentialPairN *View Name:  schematic
*****************************************************************
.SUBCKT DifferentialPairN O1 O2 VBIAS VIN VIP GND
*.PININFO GND:B O1:B O2:B VBIAS:B VIN:B VIP:B
MM2 O2 VIP net2 GND nch_mac l=30n w=100n m=1 nf=1
MM1 net2 VBIAS GND GND nch_mac l=30n w=100n m=1 nf=1
MM0 O1 VIN net2 GND nch_mac l=30n w=100n m=1 nf=1
.ENDS

*****************************************************************
*Library Name:  DATASET *Cell Name:  DifferentialPairP *View Name:  schematic
*****************************************************************
.SUBCKT DifferentialPairP O1 O2 VBIAS VIN VIP VDD
*.PININFO O1:B O2:B VBIAS:B VDD:B VIN:B VIP:B
MM1 O2 VIP net1 VDD pch_mac l=30n w=100n m=1 nf=1
MM2 net1 VBIAS VDD VDD pch_mac l=30n w=100n m=1 nf=1
MM0 O1 VIN net1 VDD pch_mac l=30n w=100n m=1 nf=1
.ENDS

*****************************************************************
*Library Name: DATASET *Cell Name:  DifferentialPairPBS *View Name:  schematic
*****************************************************************
.SUBCKT DifferentialPairP O1 O2 VBIAS VIN VIP VDD
*.PININFO O1:B O2:B VBIAS:B VDD:B VIN:B VIP:B
MM1 O2 VIP net1 O2 pch_mac l=30n w=100n m=1 nf=1
MM2 net1 VBIAS VDD VDD pch_mac l=30n w=100n m=1 nf=1
MM0 O1 VIN net1 O1 pch_mac l=30n w=100n m=1 nf=1
.ENDS
```

```
****************************************************************************
*Library   Name:    DATASET   *Cell   Name:     R  *View   Name:     schematic
****************************************************************************
.SUBCKT R O1 O2
*.PININFO O1:B O2:B
XR0 O1 O2 rnodwo l=10u w=2u m=1
.ENDS

****************************************************************************
*Library   Name:    DATASET   *Cell   Name:     C  *View   Name:     schematic
****************************************************************************
.SUBCKT C O1 O2
*.PININFO O1:B O2:B
XC0 O1 O2 cfmom_2t nr=24 lr=1u w=50n s=50n stm=1 spm=3 m=1
.ENDS

****************************************************************************
*Library   Name:   DATASET   *Cell   Name:   Cap_Feadback   *View   Name:   schematic
****************************************************************************
.SUBCKT C Vout Vin mid
*.PININFO Vout:B Vin:B mid:B
XC0 Vout mid cfmom_2t nr=24 lr=1u w=50n s=50n stm=1 spm=3 m=1
XC1 Vin mid cfmom_2t nr=24 lr=1u w=50n s=50n stm=1 spm=3 m=1
.ENDS
```

## A.3  DATASET EXAMPLE

### Synthetic Dataset Example

**[Task1 Example]**
**User Query:**
Input Stage: 1
Input blocks: DifferentialPairPBS
Other blocks:
Max blocks number: 4

**[Task2 Example]**
**User Query:**
Input Stage: 2
Input blocks: CommonSourceP DifferentialPairN
Other blocks:
Max blocks number: 4

**[Task3 Example]**
**User Query:**
Input Stage: 3
Input blocks: CommonSourceP DifferentialPairP CommonSourceP

Other blocks:
Max blocks number: 8

**[Task4 Example]**
**User Query:**
Input Stage: 1
Input blocks: CommonSourceP
Other blocks: CurrentMirrorCSN$_B$
$Max blocks number : 5$

**[Task5 Example]**
**User Query:**
Input Stage: 2
Input blocks: DifferentialPairP DifferentialPairN
Other blocks: CascodeStagePPair
Max blocks number: 7

**[Task6 Example]**
**User Query:**
Input Stage: 3
Input blocks: DifferentialPairN CommonSourceP DifferentialPairP
Other blocks: CurrentMirrorP
Max blocks number: 9

## Real Dataset Example

**Eaxmple1:**

**[User Query: ]**
Stage Numbers: 1
Compensation: None
FeadBack: Type: None, FB Network: None
InputSignal1: Differential
OutputSignal1: Single-Ended
Input Type1: NMOS
Topology1: Common Source
Load1: Simple Mirror
TailBias1: Ground

**[Golden Answer: ]**
.SUBCKT S1 Vbiasn Vin Vip Vout VDD GND
XI0 net01 Vout VDD / CurrentMirrorP
XI1 net01 Vout Vbiasn Vin Vip GND / DifferentialPairN

.ENDS

**Eaxmple2:**

**[User Query: ]**
Stage Numbers: 1
Compensation: None
FeadBack: Type: None, FB Network: None
InputSignal1: Differential
OutputSignal1: Differential
Input Type1: NMOS
Topology1: Telescopic
Load1: Wide-Swing Mirror
TailBias1: Simple Mirror

**[Golden Answer: ]**
.SUBCKT S2 Vcascp Vcascn Vin Vip Voutn Voutp Vbiasp VDD GND
XI0 Voutn Voutp VDD Vcascp Vbiasp VDD / CurrentMirrorCSPB
XI1 Voutn net01 Voutp net02 Vcascn GND / CascodeStageNPair
XI2 net01 net02 Vbiasn Vin Vip GND / DifferentialPairN
.ENDS

**Eaxmple3:**

**[User Query: ]**
Stage Numbers: 2
Compensation: Ahuja
FeedBack: Type: None, FB Network: None
InputSignal1: Differential
OutputSignal1: Single-Ended
Input Type1: Pmos(B2S)
Topology1: Folded
Load1: Wide-Swing Morror
TailBias1: Simple Mirror
InputSignal2: Single-Ended
OutputSignal2: Single-Ended
Input Type2: Pmos
Topology2: CS
Load2: Simple Mirror
TailBias2: Ground

**[Golden Answer: ]**
.SUBCKT S3 VDD GND Vin Vip Vout Vbiasn Vb1 Vbiasp Vcascn
XI0 net1 net2 VDD / CurrentMirrorP
XI1 net1 net3 net2 net4 Vb1 GND / CascodeStageNPair

```
XI2 net3 GND net4 GND Vbiasn GND / CascodeStageNPair
XI3 net7 net8 Vbiasp Vin Vip VDD / DifferentialPairPBS
XI4 net5 net6 net2 net7 Vcascn GND / CascodeStageNPair
XI5 net5 net2 VDD / CurrentMirrorP
XI6 net7 net8 GND / CurrentMirrorN
XI7 net4 Vout / C
XI8 Vout net2 VDD / CurrentSourceP
XI9 Vout Vbiasn GND / CommonSourceN
.ENDS
```

