# OpenReview forum: "ANALOGXPERT: AUTOMATING ANALOG TOPOLOGY SYNTHESIS BY INCORPORATING CIRCUIT DESIGN EXPERTISE INTO LARGE LANGUAGE MODELS"
_ICLR.cc/2025/Conference — ICLR 2025 Conference Withdrawn Submission_

### Official Review · Reviewer_RPaf · 2024-11-01

**Soundness:** 2
**Presentation:** 1
**Contribution:** 2
**Rating:** 3
**Confidence:** 3

**Summary:**

This paper introduces "AnalogXpert," a novel approach leveraging large language models (LLMs) to automate the synthesis of analog circuit topologies. The authors aim to bridge the gap between theoretical LLM capabilities and practical analog design needs by embedding circuit design expertise into the model.

**Key Focus and Problem Addressed**

The core focus of the paper is automating the complex process of analog topology synthesis, a critical component in analog circuit design. Traditional LLM-based methods fall short in practical applications as they often rely on vague design requirements and output idealized models. AnalogXpert tackles this by focusing on detailed structural requirements and device-level models, making it more applicable to real-world scenarios.

**Strengths:**

AnalogXpert innovates by representing analog topologies as SPICE code and utilizing a subcircuit library to streamline the design process, akin to the strategies employed by seasoned designers. The problem is broken down into two main tasks: block selection and block connection. This is achieved using Chain-of-Thought (CoT) and in-context learning techniques, which emulate the practical design process. Additionally, a proofreading strategy is introduced, allowing the model to iteratively refine initial designs, mirroring the iterative nature of human design processes.

**Weaknesses:**

1. The paper has poor presentation, with numerous spelling errors throughout the text, such as "Abliation Study" instead of "Ablation study" and "Feadback" instead of "Feedback". There are at least five such errors in the paper.

2. The evaluation metrics are too trivial. Since the dataset used in the paper is not open-source, the metrics are self-defined, and the baseline results are selectively chosen, it is difficult to assess the validity of the experiments.

3. The paper lacks detailed information about the analog design cases. The statement "here are approximately 60 different analog design topologies, and we select the most representative 30 analog topologies as the real data benchmark. The synthetic data benchmark is built by a random generation Python code leveraging the subcircuit library. Each synthetic data consists of four parts, the stage number, the input blocks, other given blocks, and the maximum number of blocks" is too vague, and the specific analog basic units, functionalities, and complexity levels are not clearly presented.

4. The experimental results do not conclusively demonstrate the superiority of AnalogXpert, as the low correct ratio of GPT-4o may be due to the lack of SPICE data, and the AnalogXpert's results do not achieve a very high ratio, which still poses challenges to the paper's practical applicability.

**Questions:**

1. Can the paper provide more detailed information about the analog design cases, including the specific analog basic units, functionalities, and more general complexity levels instead of stages?

2. How to handle the sizing and functionality correctness by using ANALOGXPERT?

---

### Official Review · Reviewer_k8rv · 2024-11-03

**Soundness:** 2
**Presentation:** 2
**Contribution:** 2
**Rating:** 3
**Confidence:** 5

**Summary:**

This paper introduces an LLM-based agent, AnalogXpert, to automatically generate analog circuit topologies with multiple-step reasoning.
Evaluations show that AnalogXpert achieves significant accuracy improvement in topology generation than SOTA LLMs, i.e., GPT-4o.

**Strengths:**

The paper decomposes the generation of analog circuit topology into two steps, sub-circuit selection, and sub-circuit connection, which follow the human designer steps. This strategy is novel as compared to previous LLM-based works.

The paper also develops a sub-circuit library and leverages human-based proofreading to facilitate the generation of analog circuit topologies.

An ablation study is also performed to show the impact of in-context learning and human-based proofreading on the accuracy of topology generation.

**Weaknesses:**

The other technical contributions appear limited. The Spice code generation for topology design is similar to previous methods, AnalogCoder Lai et al. (a) (24), which leverages PSpice code generation.

The paper does not interpret previous methods well. Using LLM to generate analog circuit topology is still new. It is unclear which representation is significantly better than the other, i.e., graph generation vs. code (PSpice and Spice) generation. Claiming Spice code generation is better is questionable.

The paper does not show comparisons with previous works. The reviewer does not agree “For other work related to topology synthesis, as they handle different problems with us, the comparison with them is infeasible and thus is excluded”. The previous works such as AnalogCoder Lai et al. (a) (24),  CKtGNN Dong et al., LaMAGIC Chang et al., and  RLATS Zhao & Zhang (b;c), all address topology synthesis yet with different levels of constraints. The comparison is doable and appreciated.

The paper does not clearly explain what AnalogXpert can generate now. It seems to be limited to operational amplifiers.

**Questions:**

Q1: what is the backbone used by AnalogXpert and its fundamental differences from GPT-4o?

Q2: can AnalogXpert ensure the non-ambiguous generation of analog circuit topologies that other methods suffer from?

Q3: AnalogXpert still cannot achieve more than 50% design accuracy. What is the motivation to use prompt engineering to study the capabilities of LLM in generating analog circuit topologies?

---

### Official Review · Reviewer_xjGq · 2024-11-03

**Soundness:** 2
**Presentation:** 3
**Contribution:** 2
**Rating:** 5
**Confidence:** 5

**Summary:**

This paper proposes AnalogXpert, an LLM-based agent aiming at solving practical topology synthesis problems at the sub-block level by using prompt engineering and a group of pre-defined design libraries. This proposes a benchmark containing both real data (30) and synthetic data (2k).

**Strengths:**

AnalogXpert can generate the final topology from the subcircuit level rather than the device level, which not only aligns with human design practices but also greatly reduces the length of the model output. AnalogXpert achieves better design success rates compared to GPT-4o.

**Weaknesses:**

• Benchmark is predominantly with synthetic data.
• Quantify or numerical comparisons are mostly based on GPT-4o, a generic LLM, without any other prior methods that focus on topology generation.
• The design library approach had been proven inefficient 20 years ago. As I quote from (G.G.E. Gielen and R.A. Rutenbar, 2000)https://ieeexplore.ieee.org/document/899053: “The use of a library of carefully selected analog standard cells can be advantageous for certain applications, but is in general inefficient and insufficient. Due to the large variety and range of circuit specifications for different applications, any library will only have a partial coverage for each application, or it will result in an excess power and/or area consumption that may not be acceptable for given applications. Many high-performance applications require an optimal design solution for the analog circuits in terms of power, area, and overall performance. A library-based approach would require an uneconomically large collection of infrequently used cells. Instead, analog circuits are better custom tailored toward each specific application and tools should be available to support this. In addition, the porting of the library cells whenever the process changes is a serious effort, that would also require a set of tools to automate.” How does combining LLM and the design library resolve this fundamental issue?

**Questions:**

• Can the design library be reused in other circuit types?
• What are the costs of extending the design library?

---

### Official Review · Reviewer_PLiF · 2024-11-04

**Soundness:** 2
**Presentation:** 2
**Contribution:** 2
**Rating:** 5
**Confidence:** 4

**Summary:**

AnalogXpert is a prompting strategy which aims to improve generation of analog designs by LLMs, focusing specifically on SPICE code and more closely mirroring the design steps of (human) analog-design professionals.

This strategy includes:

1. Enter a text-formatted design specifications
2. Enter a text-formatted subcircuit library
3. Begin block selection
4. Begin block connection
5. Engage in proofreading and iteration

A new benchmark is contributed consisting of real and synthetic designs.

**Strengths:**

# Strengths

1. CoT-Like Strategy More Closely Mirroring the Design Process

Via explicitly adding sub-steps, the AnalogXpert prompting strategy allows the model to more closely resemble the thinking process and workflow of designers who utilize SPICE and other tools for analog circuit design.

2. SPICE vs Python representation

As opposed to AnalogCoder which targets python representation, by AnalogXpert targeting SPICE (far preferred over python in the industry) this is poised for better immediate adoption by Analog Designers.

3. Improved Context length via utilization of sub-circuits (a more compact and higher level of circuit abstraction)

AnalogXpert keeping subcircuits as more abstracted components indeed should allow for better context lengths.

**Weaknesses:**

# Feedback:

1. Clarify earlier in paper that AnalogXpert base model used is GPT-4o, for benchmarks.
2. Clarify/Illustrate "pure GPT-4o" prompt and full comparison AnalogXpert in Appendix
3. Address Typos and Grammatical Ambiguities

## 1. Clarifying AnalogXpert base model

It is unclear that AnalogXpert is specifically a prompting-strategy + GPT-4o. The first instance I found that AnalogXpert is specifically _GPT-4o_ with the prompting strategy (as opposed to GPT-3.5 or another LLM), is on 380 page 8.  Mentioning this earlier (abstract or introduction) will really help clarify this for the reader.

## 2. Comparative Prompt from "pure GPT-4o"

It is unclear what the "pure GPT-4o" prompt is for comparison, adding the non AnalogXpert prompt for GPT-4o will be necessary for understanding the results table.

Further adding a specific example of prompts and generated answers of AnalogXpert vs GPT-4o in the appendix would be especially helpful for illustrating how the results differ.

## 3. Typos and Grammatical Ambiguities

There are several spelling and grammatical errors in this paper which will effect GPT-4o's tokenization of the prompts and understanding of the task.

### 3a) Typos Which May Affect Prompt Effectiveness

Line 923 "Userquerry:" will be 3 tokens "User-qu-erry" instead of 2 tokens "User-query" when tokenized by gpt-4o:
(Please see https://tiktokenizer.vercel.app/)

While GPT-4o may be strong enough to see past these typos, it still calls to question if the typos will affect the result quality.

### 3b) Space Consistencies which may affect tokenization

"1.Block" vs "1. Block" on line 176 would result in different tokens as well. Recommending sticking with having a convention, and including 1 space between "." and "Block" which could help the model by making it less difficult to see this section as an ordered list.

###  3c) Grammatical Ambiguities Which May Affect Prompt Interpretation

In particular, some grammatical errors may create ambiguity in the meaning of the sentence, such as line 165 step 3:

"To generate the analog circuits better follow the design steps:", which has ambiguity based on reader pause:
a, "To generate the analog circuits better, follow the design steps:" means to improve "the generation of analog circuits".
b. "To generate the analog circuits, better follow the design steps:" where better here draws attention to pitfalls of not following design steps (better in it's comparative form, e.g. "better this than that").

**Questions:**

What is the prompt used for "Pure GPT-4o?"

Being a prompting-strategy, could you give an example the reader can use to replicate results?

Would it be possible to re-run the benchmarks again with the typos, spacing, and grammar fixes?

---

### Note · Authors · 2024-11-18

I have read and agree with the venue's withdrawal policy on behalf of myself and my co-authors.